# Widespread 2013-2020 decreases and reduction challenges of organic aerosol in China

Qi Chen [1,2,8] ✉, Ruqian Miao [1,8], Guannan Geng [3,8], Manish Shrivastava [4], Xu Dao[5], Bingye Xu[6], Jiaqi Sun[5], Xian Zhang[5], Mingyuan Liu[5], Guigang Tang[5], Qian Tang[6], Hanwen Hu[3], Ru-Jin Huang [7], Hao Wang[1], Yan Zheng[1], Yue Qin [1,2], Song Guo [1,2], Min Hu[1,2] & Tong Zhu [1,2]

High concentrations of organic aerosol (OA) occur in Asian countries, leading to great health burdens. Clean air actions have resulted in significant emission reductions of air pollutants in China. However, long-term nation-wide trends in OA and their causes remain unknown. Here, we present both observational and model evidence demonstrating widespread decreases with a greater reduction in primary OA than in secondary OA (SOA) in China during the period of 2013 to 2020. Most of the decline is attributed to reduced residential fuel burning while the interannual variability in SOA may have been driven by meteorological variations. We find contrasting effects of reducing $NO_x$ and $SO_2$ on SOA production which may have led to slight overall increases in SOA. Our findings highlight the importance of clean energy replacements in multiple sectors on achieving air-quality targets because of high OA precursor emissions and fluctuating chemical and meteorological conditions.

Long-term exposure to ambient particles that have an aerodynamic diameter of 2.5 μm or smaller ($PM_{2.5}$) is associated with millions of global premature deaths per year[1]. Clean air actions are widely taken to mitigate air pollution from $PM_{2.5}$, leading to significant air-quality improvements in North America, Europe, and East Asia[2]. Remarkably, the emissions of $SO_2$ and $NO_x$ were reduced by 70% and 28%, respectively, from 2013 to 2020 in China under two key policies, namely the Air Pollution Prevention and Control Action Plan (2013–2017) and the Three-year Blue-sky Action Plan (2018–2020)[3]. The mean $PM_{2.5}$ concentrations of annual averages for 74 key Chinese cities decreased from 72 μg m$^{-3}$ in 2013 to 34 μg m$^{-3}$ in 2020[4]. Despite these advances, the $PM_{2.5}$ concentrations in most countries still far exceed the World

Health Organization (WHO) new guideline of annual mean of 5 μg m$^{-3}$ (see ref. 2). Further reduction of $PM_{2.5}$ is challenged by high fraction of organic aerosol (OA) that has an issue of a great complexity but a limited understanding of its precursors and secondary formation pathways[5,6].

The OA precursors may remain in the particle phase after being emitted to form primary OA (POA) or be oxidized in the atmosphere to form secondary OA (SOA). They consist of thousands of volatile (VOC), intermediate-volatility (IVOC), semivolatile, and low-volatility (S/LVOC) organic compounds[6]. Because of a large consumption of residential coal and biofuel and a rapid expansion of industrial activities, anthropogenic emissions of the OA precursors in

[1]State Key Joint Laboratory of Environmental Simulation and Pollution Control, Beijing Innovation Center for Engineering Science and Advanced Technology, International Joint Laboratory for Regional Pollution Control, College of Environmental Sciences and Engineering, Peking University, Beijing, China. [2]Institute of Carbon Neutrality, Peking University, Beijing, China. [3]State Key Joint Laboratory of Environmental Simulation and Pollution Control, School of Environment, Tsinghua University, Beijing, China. [4]Pacific Northwest National Laboratory, Richland, WA, USA. [5]China National Environmental Monitoring Centre, Beijing, China. [6]Zhejiang Key Laboratory of Ecological and Environmental Monitoring, Forewarning and Quality Control, Zhejiang Province Environment Monitoring Centre, Hangzhou, China. [7]State Key Laboratory of Loess and Quaternary Geology, Center for Excellence in Quaternary Science and Global Change, Institute of Earth Environment, Chinese Academy of Sciences, Xi'an, China. [8]These authors contributed equally: Qi Chen, Ruqian Miao, Guannan Geng. ✉e-mail: qichenpku@pku.edu.cn

China and India have become several times higher than those in developed countries[7–9]. Correspondingly, SOA campaign-mean concentrations reached 30–40 µg m$^{-3}$ in Chinese and Indian cities[10–12]. Although China's 2013–2020 pollution-control efforts led to significant emission reductions, rare studies have examined nationwide long-term trends in OA. Responses of OA concentrations to precursor emissions can be quite complicated due to the nonlinearity of gas-particle partitioning and secondary processes[4]. Meteorological variations and changes of biomass burning and biogenic emissions may cause extra interannual variations of OA[13–15]. Changes of anthropogenic emissions of $SO_2$, $NO_x$, and OA precursors not only affect anthropogenic OA but also particle acidity and gas-particle partitioning and consequently biogenic SOA formation[16–18]. The lack of understanding of the mitigation effectiveness, meteorological impacts, and anthropogenic–biogenic interactions associated with such high loadings of anthropogenic OA hinders the developments of effective control measures in future.

Here, we combine comprehensive surface measurement analysis and state-of-the-art OA simulations to elucidate the long-term variations of OA and their causes in China. We first used ambient observations to discover the OA trends during 2013–2021. We then simulated the OA concentrations with a revised, nested version of global chemical transport model driven by a newly developed emission inventory of OA precursors over the full-volatility range. Model results were validated by observations. A series of model sensitivity runs were conducted to examine the impacts of various drivers on OA during the action-plan period, such as changes in precursor emissions, variabilities in meteorological conditions, control of other pollutants, and changes in natural emission sources. Our results highlight the challenges of controlling OA pollution to reduce the health burden in developing countries, which may aid the acceleration of policy developments on clean-energy transition and technological innovations in the industry.

## Results

### Observed decreases of OA in China

Long-term continuous measurements of particulate organic carbon (OC) have been conducted in a recently developed national monitoring network for $PM_{2.5}$ chemical composition in North China Plain (NCP) and provincial-level monitoring sites in Yangzi River Delta (YRD) since 2015. Measurements in earlier years are only available at some research sites. As shown in Fig. 1a, c, the annual mean OC concentrations decreased by 46% in NCP from 2017 to 2021 and 36% in YRD from 2015 to 2021. Meanwhile, annual $PM_{2.5}$ concentrations decreased by about 30% in the two regions[4]. Relatively greater OC concentrations present in 2019, which deviates slightly from the general declines. The NCP data further indicate that annual OC concentrations decreased most in winter among the four seasons (Supplementary Fig. S1).

To explore the changes in POA versus SOA, we examined the OA source apportionment results obtained from aerosol mass spectrometer (AMS) measurements in 162 field campaigns across China (Supplementary Table S1 and Fig. 1b). Each campaign covers a short period of 2 weeks to 3 months. NCP has the highest regional-mean OA concentration of 23.1 µg m$^{-3}$ in contrast to the region-mean concentrations of 13.5–15.5 µg m$^{-3}$ in YRD, Northwest (NW), and Pearl River Delta (PRD) (Supplementary Table S2). POA accounts for about 50% of OA in north regions (NCP and NW) and about 40% in south regions (YRD and PRD), reflecting a spatial difference in sources[19]. The data in five cities from different regions in China indicate clear OA declines in winter, accompanied with increasing SOA mass fractions in three of the cities (Fig. 1e, f). In Beijing, where the campaign data covers over 50% of the period of 2013–2020, the declines of OA and the increases of SOA fractions present in all seasons, with more significant changes in winter. Moreover, all-campaign-mean OA concentrations decreased from 20.6 to 9.0 µg m$^{-3}$, and SOA mass fractions increased from 53 to 69% (Supplementary Table S3). To conclude, the observations suggest

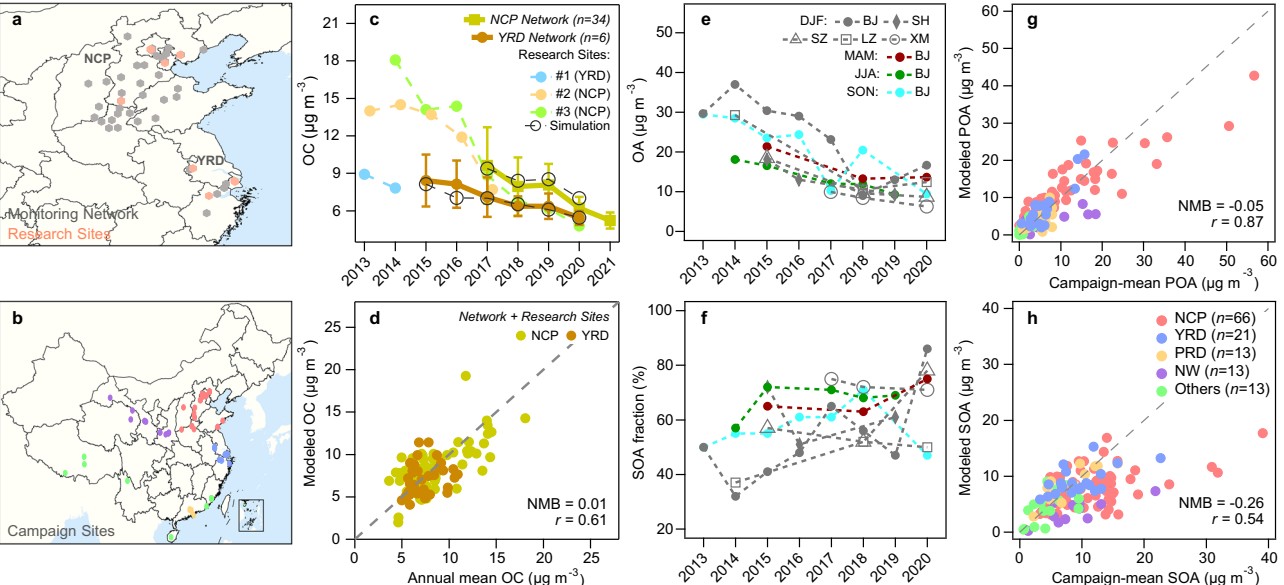

**Fig. 1 | Long-term observations of organic aerosol (OA) in China and their comparisons to model simulations. a** Locations of the monitoring sites for organic carbon (OC) including 34 sites in a recently developed national monitoring network for $PM_{2.5}$ chemical composition in North China Plain (NCP), 6 provincial monitoring sites in Yangzi River Delta (YRD), and 3 research sites in the two regions. **b** Locations of the campaign sites for aerosol mass spectrometers. **c** Annual mean OC concentrations from the continuous measurements of long-term monitoring sites. Error bars represent the standard deviations (NCP) and the ranges (YRD) of the annual concentrations across sites. **d** Annual mean OC concentrations at each site compared to the modeled OC. **e, f** Campaign-mean OA concentrations and

secondary OA (SOA) mass fractions in OA obtained from individual campaigns in different seasons (DJF: winter; MAM: spring; JJA: summer; SON: fall) during 2013–2020 in cities from five main regions in China (BJ: Beijing in NCP; SH: Shanghai in YRD; SZ: Shenzhen in PRD; LZ: Lanzhou in Northwest (NW); XM: Xiamen in southeast coast). **g, h** Campaign-mean primary OA (POA) and SOA concentrations compared to the modeled concentrations. Source data are provided as a Source Data file. The coastline boundaries in the map are originated from Natural Earth free vector map data (https://www.naturalearthdata.com/). The administration boundaries are originated from the National Earth System Science Data Center(https://www.geodata.cn).

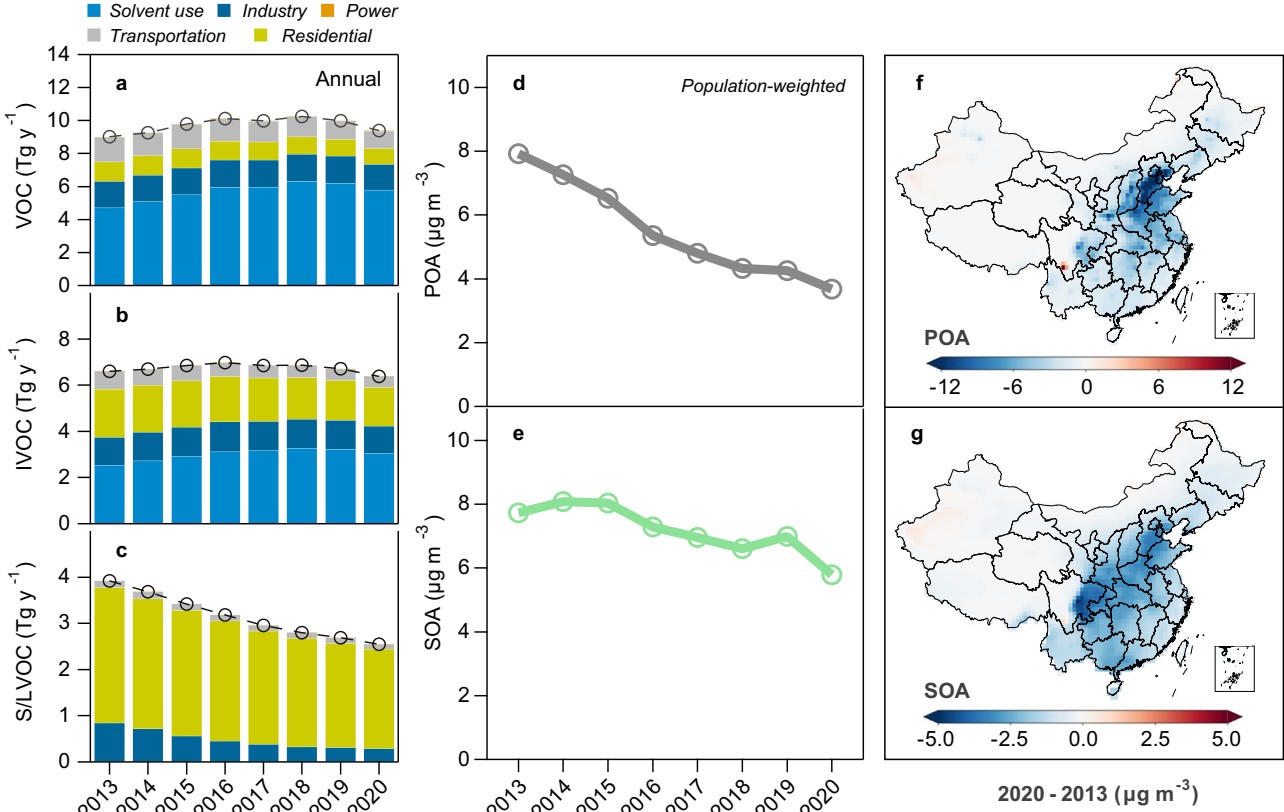

**Fig. 2 | Simulated organic aerosol (OA) precursors, primary OA (POA), and secondary OA (SOA) trends in China. a–c** Anthropogenic emissions of OA precursors from different source sectors. Among volatile organic compounds (VOC), only aromatic compounds, glyoxal, methylglyoxal, isoprene, and terpenes are considered as OA precursors. **d**, **e** Simulated population-weighted annual mean concentrations of POA and SOA over the action-plan period. **f**, **g** Spatial difference of simulated POA and SOA mean concentrations between 2020 and 2013. Source data are provided as a Source Data file. The coastline boundaries in the map are originated from Natural Earth free vector map data (https://www.naturalearthdata.com/). The administration boundaries are originated from the National Earth System Science Data Center(https://www.geodata.cn).

widespread decreasing OA concentrations with increasing SOA mass fractions in China.

## Drivers of interannual variations of POA and SOA

The decline of OA follows the emission reduction of OA precursors. We developed VOC, IVOC, and SVOC emission inventories for China that reflect the changes of fuel uses and end-of-pipe control measures from 2013 to 2020 (Fig. 2a–c). Anthropogenic emissions of S/LVOC are estimated to be $2.6–4.0\,Tg\,y^{-1}$, among which about $1.4–2.5\,Tg\,y^{-1}$ are emitted as POA. The S/LVOC emissions decreased by 35% from 2013 to 2020, and 70% of the reduction occurred during the 2013–2017 period. The replacement of residential stoves with natural gas and electric stoves contributes 56% of the S/LVOC reduction, especially in winter (Supplementary Fig. S2)[3,20]. The estimated anthropogenic IVOC emissions are $6.5–7.2\,Tg\,y^{-1}$, showing an 8% decrease from 2013 to 2020. Rapid expansion in solvent use leads to high emissions of volatile chemical products that offset the reductions in residential and transportation emissions[8,21]. In addition, about 35% of non-methane VOC emissions (mostly aromatic compounds) may act as OA precursors (i.e., $9.0–10.3\,Tg\,y^{-1}$). The high aromatic fraction in China is explained by the rapid increase in solvent use and industrial activities since 2000[8]. The economic slowdown during the COVID-19 outbreak led to a significant decrease of anthropogenic emissions in 2020. Overall, our estimated annual emissions of IVOC and S/LVOC are comparable with other nationwide inventory values with relative differences of 11–36% (Supplementary Fig. S3)[7,22–24]. Compared to the emissions in the United States[9], the annual emissions of IVOC and S/LVOC in China are about 2 times

higher but with much less contributions from open biomass burning (Supplementary Fig. S4).

We applied these emissions in GEOS-Chem and conducted simulations under assimilated meteorology with key model updates that improve the model performance on $PM_{2.5}$ chemical components and surface hydroxyl radical (OH) concentrations[19]. The model can reproduce the decreasing trends in OC in NCP and YRD, and the simulated OC concentrations agree well with the observations (Supplementary Fig. S5). For all-site comparisons, the normalized mean bias (NMB) is 0.01 and the Pearson correlation coefficient ($r$) is 0.61 (Fig. 1d). Furthermore, model simulations reasonably capture the observed POA (NMB = −0.05; $r = 0.87$) and SOA (NMB = −0.26; $r = 0.54$), and the comparisons do not indicate significant regional or seasonal model biases (Fig. 1g, h and Supplementary Fig. S5). Statistical values are similar for the main regions of NCP, YRD, and PRD (Supplementary Table S2). Note that yearly model comparisons to campaign observations are statistically less meaningful, showing relatively greater NMBs for smaller sample sizes (Supplementary Table S3). On average, S/LVOC and IVOC together contribute to about 75% of the simulated SOA in China. Such a high contribution is consistent with the findings from ambient SOA potential analysis and other modeling work in regions of high anthropogenic emissions[25–28].

Figure 2d, e shows the simulated annual mean concentrations of POA and SOA in China. The population-weighted POA concentrations exhibit a steady decline of 53% from $7.9\,\mu g\,m^{-3}$ in 2013 to $3.7\,\mu g\,m^{-3}$ in 2020, while SOA shows a less reduction of 25% from 7.7 to $5.8\,\mu g\,m^{-3}$. Spatially, the reduction in POA mainly occurs in NCP where the residential emissions are high (Fig. 2f, g), which is consistent with the

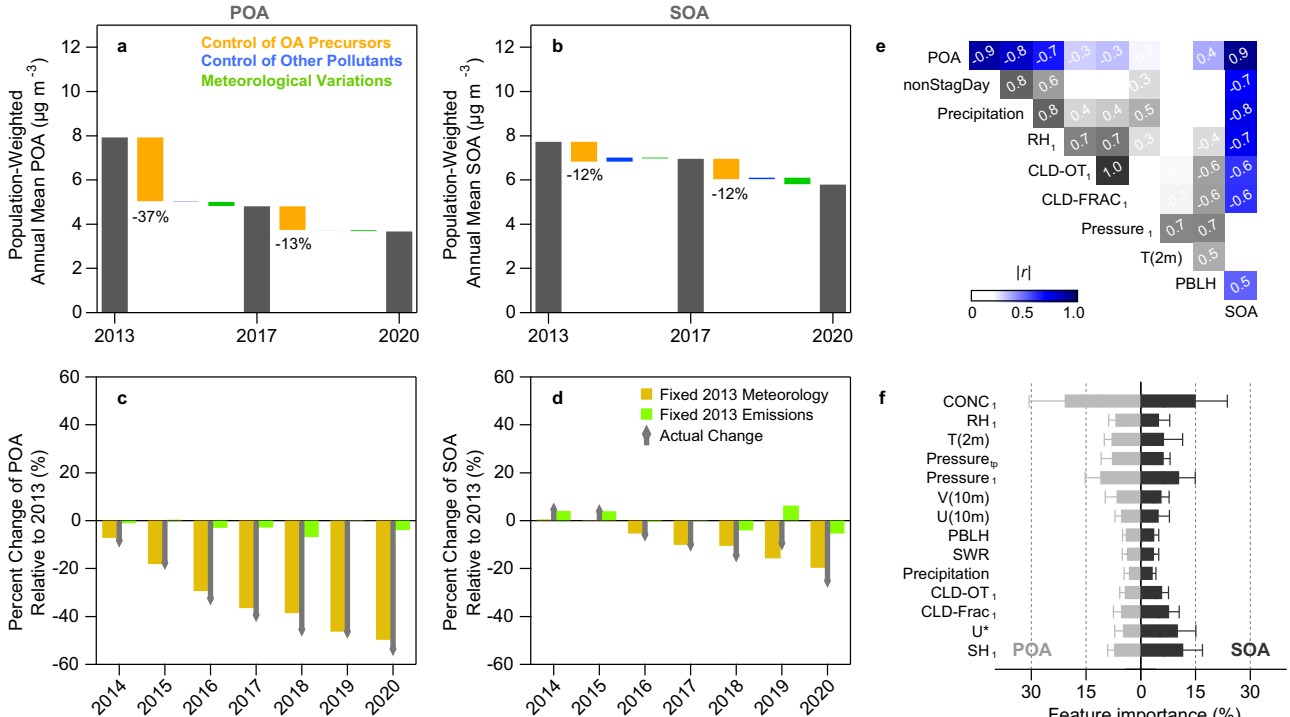

**Fig. 3 | Drivers of primary organic aerosol (POA) and secondary organic aerosol (SOA) trends in China. a**, **b** Emission- and meteorology-driven variations of annual population-weighted POA and SOA in China during the two action-plan periods. Percent values represent the POA or SOA changes relative to 2013. **c**, **d** Yearly variations of annual population-weighted POA and SOA under fixed-2013-emission or fixed-2013-meteorology scenarios compared to their base-model simulated changes relative to 2013. **e** Pearson *r* values for the correlations between normalized annual means of key meteorological parameters and those of POA and SOA simulated under the fixed-2013-emission scenario. **f** Feature importance of meteorological inputs to the meteorology-driven interannual variations of POA and SOA resolved by random forest (RF) model analysis. CONC stands for the concentrations of POA and SOA used in RF which are the model outputs under the fixed-2013-emission scenario. PBLH, SWR, T(2 m), U(10 m), and V(10 m) represent planetary boundary layer height, surface shortwave radiation, air temperature at 2 m, the east–west and north–south wind components at 10 m, respectively. The subscripts, 1 and tp, denote the first model layer and tropopause, respectively.

greater OA decline in Beijing than in other cities (Fig. 1e). By contrast, the decrease of SOA is widespread because of atmospheric processing of diverse precursors despite of the scattered changes in precursor emission (Supplementary Fig. S6). For a particular case in southern Sichuan, intensive forest fires in 2020 cause significant enhancements in POA and SOA (Supplementary Fig. S7). Nationally, POA contributes to two-thirds of the OA decline from 2013 to 2020. The mass fraction of SOA in OA becomes greater in most regions (Supplementary Fig. S8), and the average fraction of SOA increases from 49 to 61%, which are consistent with the AMS observations (Fig. 1e, f and Supplementary Table S3).

Under fixed meteorological inputs or emissions, the model results further indicate that the emission changes in OA precursors is the predominant factor to lower OA concentrations from 2013 to 2020, especially during the first phase of 2013–2017 (Fig. 3a, b). The annual reduction of POA emissions relative to 2013 is about 3–9%, corresponding to 1–15% of annual decline in POA concentrations (Supplementary Fig. S9). Meteorological variations may have led to additional decreases of annual mean POA by 0–6% (Fig. 3c). Unlike POA, both positive and negative annual changes of SOA present when changes in SOA precursor emissions are limited (Supplementary Fig. S9). The modeled meteorology-driven changes of annual mean SOA range from −5 to 6% (Fig. 3d), which explains the majority of the year-to-year variations of SOA on top of the emission-driven decline from 2013 to 2020. The year of 2019 appears to be most meteorologically unfavorable for reducing OA, which agrees with the observed high OC concentrations in NCP and YRD (Fig. 1c). Compared to anthropogenic emissions, natural changes in wildfires and biogenic emissions play negligible roles on OA interannual variability (Supplementary Fig. S10).

To identify key meteorological parameters that affect OA, we examined the correlations between the interannual variations of these parameters and the meteorology-driven annual changes of POA and SOA that are simulated under the fixed-2013-emission scenario (Supplementary Fig. S11). Non-stagnation days (nonStagDay), precipitation, and relative humidity (RH) correlate strongly and negatively (*r* = −0.7 to −0.9) with POA and SOA, explained by the influences of atmospheric dilution and removal on aerosol concentrations (Fig. 3e). SOA correlates better with cloud fraction (CLD-FRAC) and cloud optical thickness (CLD-OT) (*r* = −0.6) than POA does (r = −0.3), suggesting that cloud-driven radiation changes may significantly affect the photochemical oxidation of OA precursors to form SOA[29].

To consider the spatial difference of meteorological influences on surface PM$_{2.5}$ concentrations[30], we trained a random forest model with assimilated meteorological inputs to predict the meteorology-driven interannual variations of POA and SOA in each grid (Supplementary Note S1 and Supplementary Fig. S12). The feature importance and SHapley Additive exPlanations (SHAP) values confirm that CLD-FRAC and CLD-OT are more relevant and important parameters for explaining the interannual variations in SOA than in POA (Fig. 3f and Supplementary Fig. S13). Two other parameters, specific humidity (SH) and friction velocity (U*), are more important for SOA than for POA. These parameters are used for the calculations of air density and gaseous dry deposition (affecting SOA precursors) in the model, respectively.

Atmospheric reaction conditions also affect the SOA formation. Interestingly, the reductions of SO$_2$ and NO$_x$ during 2013–2020 play opposite roles in affecting SOA production. Supplementary Fig. S14 shows greater modeled concentrations of atmospheric oxidants (i.e.,

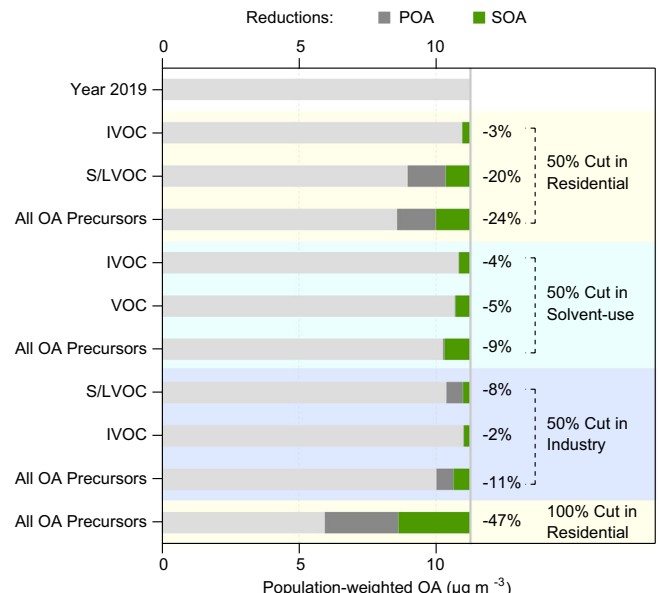

**Fig. 4 | Response of organic aerosol (OA) on precursor reductions.** Cutting all OA precursors in each sector represent model scenarios of clean-energy transition. Cutting individual category of OA precursors in each sector represent model scenarios of end-of-pipe control. OA concentrations are population-weighted annual mean concentrations. Percent values represent the reductions relative to the 2019 concentrations.

OH and ozone) in 2020 than in 2013 in northern and eastern China and Sichuan Basin. The $NO_x$ reduction is one of the main drivers of the elevated oxidant levels[31] and enhances SOA production (Supplementary Fig. S15). By contrast, the $SO_2$ reduction leads to significantly lower aerosol liquid water content (ALWC) and greater aerosol pH in those regions, affecting aqueous SOA production from small carbonyls (e.g., glyoxal) and isoprene epoxydiols[17,32]. These changes offset each other among regions, leading to only 0–3% of net increases in annual mean SOA (Fig. 3a, b and Supplementary Fig. S15). On a regional scale and by monthly evaluation, we found much greater OA changes that can be attributed to the meteorological and chemical drivers (Supplementary Fig. S16). However, the model skills in describing finer temporal and spatial variations of OA might be limited because of large uncertainties in estimated emissions and the lack of observational constraints.

**Response of OA on precursor reductions**
Figure 4 shows the model results under extreme emission-cut scenarios for anthropogenic OA precursors from different sectors in China. We consider the scenarios of cutting all OA precursors in each sector as clean-energy transition pathways and the scenarios of cutting individual category of OA precursors as end-of-pipe solutions. The population-weighted annual mean concentration of OA in the base year of 2019 is 11.2 µg m$^{-3}$ in which 38% is POA. Because the majority of S/LVOC is contributed by residential sources (Fig. 2c), a full elimination of residential organic emissions may greatly reduce POA by 63% and SOA by 38%, leading to a total decline of 47% of OA. Less reduction in SOA is expected because residential is the predominant source sector of POA whereas solvent use and industry also contribute substantially to SOA. The 100% cut in residential is not a practical solution because of the large volume and dispersed nature of residential sources. A less aggressive cut of 50% of the residential emissions may lower OA by 23%. By contrast, solvent use contributes greatly to the emissions of IVOC and VOC. A 50%-less solvent use has a potential to lower OA by 9%. In addition, about 11% of OA reduction may be achieved by a 50% cut in industrial emissions of OA precursors (i.e., mainly from S/LVOC reduction). Among the source sectors, residential remains as the most

effective sector for controlling OA pollution in China. Our results suggest that a coordinated multi-sector control strategy is necessary to effectively lower OA concentrations. However, the fact that a 50% emission cut on all anthropogenic OA precursors cannot guarantee an OA level below 5 µg m$^{-3}$ indicates a great challenge for China to achieve the new WHO air-quality guideline.

## Discussion
Our study identifies a large and rapid decrease in OA from 2013 to 2020 in China from observations and model perspectives, indicating remarkable achievements of the implementation of clean air actions (especially residential fuel replacement). A similar extent of decline (25–50% of OC) occurred in the United States from 1990 to 2012[13]. Given the predominant anthropogenic contributions to high OA concentrations, variations in natural sources such as biogenic sources and wildfires negligibly affect interannual variations of OA in China. By contrast, variations in meteorological conditions play a more important role on OA. The modeled POA and SOA are both anti-correlated with ventilation, precipitation, and relative humidity, whereas SOA is also affected negatively by cloud fraction and optical thickness due to photochemical impacts, leading to a greater dependence of SOA variability on meteorology than POA. We further show the reduction of $NO_x$ and $SO_2$ in China may play opposite roles in SOA formation and have caused a small offset to the SOA reduction from 2013 to 2020. The lesser reduction in SOA than in POA that has occurred in China verifies the outcome of the lack of control on solvent-use emissions of IVOC and VOC and the chemical and meteorological impacts.

The simulation of anthropogenic OA over a full-volatility range remains highly uncertain. This study reproduces the observed OC, POA, SOA, and IVOC concentrations in China (Fig. 1 and Supplementary Note S3), indicating a good estimate of the total amounts of S/LVOC and IVOC emissions. Recent emissions from transportation are consistent among inventories (Supplementary Fig. S3)[7,22,23]. Our residential emissions are greater than others because of the use of harmonized emission factors[7,24]. Lower residential emissions, however, would result in lower simulated SOA and worse model-observation comparisons. The representation of chemical processes for SOA formation and aging in chemical transport models can be an important source of model bias[6]. Our model simulations reasonably reproduce the regional variations of OA (Supplementary Fig. S17 and Supplementary Table S2), suggesting a minor impact of simplified model representation of multigenerational production and fragmentation pathways on SOA in regions of high anthropogenic emissions[33,34]. Moreover, the meteorology-driven changes of aerosol microphysical properties (e.g., phase state and viscosity) and their impacts on SOA formation and lifetime are poorly understood and beyond the model capability[35], which is likely an important local controlling factor but not a dominant factor across all regions. Future improvements on the volatility distributions of individual source sectors and the description of the conversion and fate of full-volatility-range organics are needed to reduce the regional model bias in SOA simulations.

Our findings have three important policy implications for OA pollution control: (1) The residential sector is still the biggest contributor of OA in China. As the residential fuel replacement slows down, the OA pollution likely becomes persistently high in China and thus challenges the consideration of tightening national air-quality standards. Future policies on air-quality improvement should prioritize the continuation of residential clean-energy transition; (2) Abatements of VOC have been suggested as a near-future priority for ozone-pollution control in China[31]. Solvent use is now the largest contributor of non-methane VOC and SOA precursors. Coordinated control on ozone and SOA should encourage measures on solvent-use emissions; (3) Our model simulations suggest the $NO_x$ reduction increased SOA during 2013–2020 in China. In the longer term, if the $NO_x$-suppressed regime is transitioned to a $NO_x$-limited regime, the

$NO_x$ reduction would decrease surface OH to lower photochemical production and lower inorganic nitrate concentrations to decrease ALWC and thus aqueous SOA[36], leading to co-benefits on SOA pollution control.

Notably, high emissions of OA precursors are expected in other emerging economies such as India and Southeast Asian countries (Supplementary Fig. S18). High concentrations of OA in $PM_{2.5}$ have already been a serious problem there[10,37]. Similar considerations may be taken to develop effective pollution-control measures in those countries. To gain maximum co-benefits and ensure the effectiveness of pollution mitigation policies, future research needs to focus on (1) characterization of OA species, OA precursor class, or emission sectors by health impacts, which leads to better-tailored control strategies on the most harmful OA sources; and (2) quantification of the potential climate impacts on OA under different future emission scenarios[38].

## Methods

### Model descriptions

We use the nested GEOS-Chem model (13.3.1; https://doi.org/10.5281/zenodo.5703364) with a revised Complex SOA scheme to simulate POA and SOA at 0.5° × 0.625° horizontal resolution over the East Asia[19,39]. The model is driven by assimilated meteorology from the NASA Modern-Era Retrospective Analysis for Research and Application, Version 2 (MERRA2; http://gmao.gsfc.nasa.gov/reanalysis/MERRA-2). Anthropogenic emission inventories of OA precursors over the full-volatility range from 2013 to 2020 are developed on the basis of fuel-specified Multi-resolution Emission Inventory for China (MEIC) version 1.4[3]. Compared to the previous version of MEIC, residential biofuel and coal use are updated on the basis of nationwide on-site surveys[40]. The IVOC emissions are estimated from non-methane VOC by applying specific emission ratio and volatility distribution to individual source sectors (i.e., industry, solvent-use, transportation, power, and residential) and fuel types (i.e., diesel, gasoline, coal, residential biofuel and coal, etc.), which are harmonized from measurement-based literature results (Supplementary Table S4). We then validate the emissions and volatility distributions by comparisons to ambient observations of IVOC (Supplementary Figs. S19–S20). The S/LVOC emissions are estimated by using the common empirical S/LVOC-to-OC$_{inventory}$ method but with updated ratios and measurement-based volatility distributions for individual sectors (Supplementary Table S5). About 54–63% of the emitted S/LVOC readily condense to the particle phase to form POA. The good comparisons of modeled POA and POA volatility distributions to the observations validate our estimated S/LVOC emissions (Supplementary Figs. S5 and S21). Biofuel use is treated as an anthropogenic source while open biomass burning is considered as a natural source in this study. All other emissions, model modifications and configurations are described in Supplementary Note S1. Supplementary Table S6 lists the model modifications for HONO sources, heterogeneous uptake of $HO_2$, $SO_2$, and $NO_2$, and SOA parameterizations etc. and the improvements of model performance. Updated SOA yields are provided in Supplementary Table S7.

### Simulations and scenarios

Base simulations are performed in GEOS-Chem from 2013 to 2020 with a spin-up period of one month. Chemical boundary conditions at the edges of the simulation domain are provided by global simulations at 2.5° × 2.5° horizontal resolution. Model runs under various scenarios are listed in Supplementary Table S8 and are described in detail in Supplementary Note S1. We conducted the model runs from 2014 to 2020 with fixed-2013 meteorology to test the emission-driven (all sources) interannual variations of OA. Similar model runs with fixed-2013 emissions were conducted to test the meteorology-driven

interannual variations of OA relative to 2013 (Fig. 3c, d). We then used these model results as well as assimilated meteorological inputs to train a random forest model with SHAP analysis to predict the interannual variations of OA and to identify the key parameters. For the two phases of 2013–2017 and 2017–2020, we first run the model in 2017 with 2013's emissions of anthropogenic OA precursors, 2013's emissions of anthropogenic pollutants other than OA precursors, or 2013's meteorology, and then run the model in 2020 with 2017's emissions and meteorology. This set of test is to quantify the impacts of anthropogenic emission control on OA, for which meteorology-driven differences only reflect 2017 vs. 2013 or 2017 vs. 2020 not the four-year period (Fig. 3a, b). We applied the population data from the Gridded Population of the World (https://sedac.ciesin.columbia.edu/data/collection/gpw-v4) dataset for 2015 to derive the population-weighted concentrations.

### Observations

Comprehensive data sets of ambient observations are used herein for analyzing OA trends and model evaluations. Annual mean concentrations of particulate OC from 2017 to 2021 in NCP are obtained from continuous measurements in a national network for monitoring the $PM_{2.5}$ chemical composition which currently consists of 34 sites in 28 cities. Similar OC data in YRD are obtained from continuous measurements at 6 provincial-level $PM_{2.5}$ composition monitoring sites in Zhejiang province, Shanghai, and Jiangsu province. In addition, annual OC data at 9 research sites are added to the observation dataset. The mean POA and SOA concentrations from 162 field campaigns are summarized and listed in Supplementary Table S1. These OA measurements are conducted by Aerodyne aerosol mass spectrometers or aerosol chemical speciation monitors from 2013 to 2020, and the source apportionments are analyzed by using the positive matrix factorization (PMF) method with the PMF2 or ME2 solvers[11]. The concentrations of hydrocarbon-like, cooking-related, biomass burning-related, and coal-combustion-related OA factors and those of the oxygenated OA factors are summed up to represent the POA and SOA concentrations, respectively. We also collect a large dataset of campaign-mean concentrations of total IVOC in China to validate the precursor emissions, and derive the volatility distributions of IVOC and particle-phase S/LVOC from ambient gas or particle volatility measurements in different places to validate the volatility-bin settings in the emission inventories. The model results from the same locations and measurement periods and arithmetic-mean concentrations are used for model-observation comparisons (Supplementary Figs. S19–S21). Detailed descriptions about the observations are provided in Supplementary Note S2. The choice of PMF factors analog to the modeled OA categories, the model-observation comparisons, and model consistency within various GEOS-Chem simulations are described in detail in Supplementary Note S3.

### Reporting summary

Further information on research design is available in the Nature Portfolio Reporting Summary linked to this article.

## Data availability

The modeled OA concentrations in China generated in this study are available in the public repository (https://doi.org/10.5281/zenodo.11114881). The raw monitoring network data are protected and are not available due to data privacy laws. Source data are provided with this paper.

## Code availability

The revised GEOS-Chem code, the run directory for GEOS-Chem, and the code for reproducing the figures are available at the public repository (https://doi.org/10.5281/zenodo.11114855).

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

## Acknowledgements

This work is supported by the National Natural Science Foundation of China (92044301, 42293323, 41961134034) and the 111 Center of Urban Air Pollution and Health Effects (B20009). M.S. was supported by the United States Department of Energy (DOE) Office of Science, Office of Biological and Environmental Research through the Early Career Research Program. The authors gratefully acknowledge Dr. Tao Xue and the National Institute of Health Data Science at Peking University for providing computation resources, Dr. Colette Heald for helpful discussion, Dr. Weiwei Hu for providing the volatility distribution data, and the MEIC team for their support on emission inventory developments.

## Author contributions

Q.C. designed the study. G.G., H.H., and R.M. developed the emission inventories. R.M. and H.W. performed the simulations of the chemical transport model and random forest model. Q.C., R.M., H.W., Y.Z., and R.H. analyzed the field data. X.D., B.X., J.S., X.Z., M.L., G.T., S.G., M.H., and Q.T. provided the long-term OC data. Q.C., R.M., M.S., Y.Q., and T.Z. analyzed the model data. Q.C. and R.M. wrote the paper with comments from all the other authors.

## Competing interests

The authors declare no competing interests.
