## [Peer Review File · Nature Communications]

Widespread 2013-2020 Decreases and Reduction Challenges of Organic Aerosol in ChinaREVIEWER COMMENTS

Reviewer #1 (Remarks to the Author):

Summary of Article:

This study investigates the long-term trends of organic aerosol (OA) in China, motivated by the Chinese government's two-phase program to reduce OA between 2013-2020. The researchers combine modeling and observation methods effectively. Nation-wide observations are gathered from the North China Plain and province-level sites in the Yangtze River Delta. Aerosol mass spectrometers (AMS) and aerosol chemical speciation monitors are used for data collection. The authors employ the GEOS-Chem chemical transport model in high-resolution nested-grid mode. Various configurations, including simulations with 2014-2020 emissions and fixed 2013 meteorology, enable separation of OA precursor reduction effects from meteorological influences on OA concentrations. Notably, the study explores the implications of the fixed-emissions model runs using a random forest model, identifying meteorological variables most important to OA variability, an interesting approach.

Key findings include:

1. Declines in OA are mainly due to decreased Primary OA (POA).
2. Anthropogenic emissions drive OA trends, especially for POA.
3. Meteorology significantly affects Secondary OA (SOA), with $\sim\pm 5\%$ variations due to meteorological factors.
4. Meteorological features show negative correlations of both POA and SOA with precipitation, humidity, and non-stagnant days. Cloud-related factors impact SOA more than POA due to their influence on photochemistry.
5. Reductions in SO_x and NO_x in China offset each other, resulting in negligible net changes in annual mean POA and SOA.

To further reduce OA pollution, the authors recommend:

1. Prioritize replacing solid fuels with clean fuels or electricity for residential fuel burning.
2. Encourage products with less volatile components and focus on collecting and replacing high-emission products to reduce solvent-related SOA.
3. Continue the clean energy transition and invest in new technologies.

In sum, the authors have produced an excellent piece. This paper contains discussion of a novel data record, some elegant modeling techniques, and an investigation of the success of the last decade of Chinese air quality policy. It is a worthy achievement. The findings themselves are not a surprise: the finding that POA is easier to control than SOA is consistent with the US and European experiences; it is similarly unsurprising to learn that meteorological variability is important to SOA variability. The investigation of meteorological drivers is also as predicted: OA is anti-correlated with ventilation, precipitation, and relative humidity; SOA is anti-correlated with cloud fraction and optical thickness due

to photochemical impacts. The surprise to me was that the SO_x and NO_x reductions offset one another, but the authors do not highlight these results. The work is valuable in that it is an examination of all of these findings in the Chinese context, but the findings themselves are as expected.

General issues: Model vs AMS definitions of POA and SOA

This paper primarily concerns a comparison between AMS observations and models of organic aerosol, to better understand the controls on the trends of these pollutants. These authors, like many before them, must confront an issue in making these comparisons: interpretations of AMS data do not make the same distinction between ‘primary’ and ‘secondary’ organic aerosols made by models. AMS observations are essentially apportioned between ‘oxidized’ and ‘not-oxidized’ OA, which is related to the POA/SOA distinction used by modelers but not identical to it.

The choices made by the authors in this comparison are logical: Hydrocarbon-like (HOA), cooking-related (COA), biomass-burning-related (BBOA) and coal-combustion-related OA (CCOA) is apportioned as POA, oxygenated OA factors – presumably some combination of Less-Oxidized Oxidized Organic Aerosol (LO-OOA) and More-Oxidized Oxidized Organic Aerosol (MO-OOA) – are apportioned as SOA. However, they do not agree with several different recent studies of Chinese OA (Brewer et al., 2023; Wang et al., 2021). In both of those studies, the authors found that their observations were much more consistent with a treatment of POA which apportioned GEOS-Chem’s Emitted POA (EPOA) tracers to the HOA/COA/BBOA/CCOA and oxidized POA (OPOA) tracers to the OOA alongside SOA. This disagreement actually presents an opportunity for the authors here to explain how their model and their finding fit into the existing literature.

It is not clear to me how the authors differentiated between what GEOS-Chem Classic labels EPOA and OPOA, though an examination of the cited paper by Miao et al (2021) [citation 19] seems to suggest that some portion of what would have been considered POA in the classic GEOS-Chem ‘simple’ and ‘complex’ mechanisms might now be considered SOA by the authors’ model. This would be consistent with their finding that the S/LVOC and IVOC together contribute ~75% of the simulated SOA in China (Lines 123-124), as I believe older GEOS-Chem mechanisms (Hodzic et al., 2016; Pye et al., 2010) mostly consider S/LVOC and IVOC as POA-forming. Regardless, I think the authors ought to address this in the text or supplement.

Brewer, J. F., Jacob, D. J., Jathar, S. H., He, Y., Akherati, A., Zhai, S., et al. (2023). A Scheme for Representing Aromatic Secondary Organic Aerosols in Chemical Transport Models: Application to Source Attribution of Organic Aerosols Over South Korea During the KORUS-AQ Campaign. *Journal of Geophysical Research: Atmospheres*, 128(8), e2022JD037257. <https://doi.org/10.1029/2022JD037257>

Hodzic, A., Kasibhatla, P. S., Jo, D. S., Cappa, C. D., Jimenez, J. L., Madronich, S., & Park, R. J. (2016). Rethinking the global secondary organic aerosol (SOA) budget: stronger production, faster removal, shorter lifetime. *Atmos. Chem. Phys.*, 25.

Miao, R., Chen, Q., Shrivastava, M., Chen, Y., Zhang, L., Hu, J., et al. (2021). Process-based and observation-constrained SOA simulations in China: the role of semivolatile and intermediate-volatility organic compounds and OH levels. *Atmospheric Chemistry and Physics*, 21(21), 16183–16201. <https://doi.org/10.5194/acp-21-16183-2021>

Pye, H. O. T., Chan, A. W. H., Barkley, M. P., & Seinfeld, J. H. (2010). Global modeling of organic aerosol:

the importance of reactive nitrogen (NO_x and NO₃). *Atmospheric Chemistry and Physics*, 10(22), 11261–11276. <https://doi.org/10.5194/acp-10-11261-2010>

Wang, J., Ye, J., Zhang, Q., Zhao, J., Wu, Y., Li, J., et al. (2021). Aqueous production of secondary organic aerosol from fossil-fuel emissions in winter Beijing haze. *Proceedings of the National Academy of Sciences*, 118(8), e2022179118. <https://doi.org/10.1073/pnas.2022179118>

Specific Edits:

Line 105 – I was surprised by the ‘35% of NMVOC acting as an OA precursor’ value, and went and looked through the cited work by Li et al. (2014). I did not find the source of the claimed percentage. It is entirely possible that I have missed something, and welcome being corrected, but the authors should double check the citation.

Li, M., Zhang, Q., Streets, D. G., He, K. B., Cheng, Y. F., Emmons, L. K., Huo, H., Kang, S. C., Lu, Z., Shao, M., Su, H., Yu, X., and Zhang, Y.: Mapping Asian anthropogenic emissions of non-methane volatile organic compounds to multiple chemical mechanisms, *Atmos. Chem. Phys.*, 14, 5617–5638, <https://doi.org/10.5194/acp-14-5617-2014>, 2014.

Figure 3: I really like this figure. It is both information-dense and understandable. The presentation choices in panels A-D in particular are very well done. My only complaint is with Panel F. This panel contains several confusing references: the “Conc.”, “TROPO-P”, and “P1” labels should be clarified. Two of these features (“Conc.” and “TROPO-P”) do not appear to be discussed in the supplemental materials

Line 172: inclusion of “Besides” here is confusing and out of place

Line 194: “Except the uncertainty in precursor emissions, the representation of chemical processes for the SOA formation and aging is greatly simplified in chemical transport models.” This line is confusing to me. It seems to suggest that CTMs don’t simplify the uncertainty in precursor emissions, but this is not very true, especially considering the assertion in lines 123-124 that S/LVOC and IVOC contribute ~75% of SOA in China. Possibly one could argue that the representation of precursor emissions is only ‘very simplified’ rather than ‘greatly simplified’ but I’m not sure that’s a worthwhile line to draw. I suggest rephrasing this line to make the intended statement clearer.

Line 228-229: authors suggest that the large decrease in OA over China from 2013-2020 is “unprecedented” (a typo) and then in the very next sentence suggest a precedent: “[a] similar extent of decline ... occurred in the United States from 1990-2012”. I think this is the wrong word choice.

Line 242: “solvent use is the second large contributor” should read “solvent use is the second largest contributor”.

Line 246-247: “lowering OA mass ... is essentially a thrust on clean energy transition and technology innovations.” This phrasing doesn’t make sense – I don’t know what the authors mean by ‘a thrust’.

Lines 249-260: This section is included in the ‘Policy Implications’ section of the paper but does not really

consist of any policy implications. It instead lays out two (extremely correct) needs for future research, which I summarize as:

1. Characterization of OA species (or OA precursor emissions class or emissions sector) by health impacts is urgently needed to better tailor control strategies at the most damaging OA sources.
2. More research is necessary on potential future climate impacts on OA, and how that will change policy.

These points are excellent but not policy. I assume this inclusion was made because the paper contains no 'Conclusions' section, which is where they really do belong. The authors should re-organize this section to make clear exactly what the policy recommendations are and what the attendant research is that could make those recommendations even better. I suggest a numbered list of the three real policy implications in this section, followed by this additional discussion of necessary future directions.

Line 275: "24% of the emitted carbon present in low-volatility bins with saturation concentrations (C^*) of $\leq 1 \mu\text{g m}^{-3}$ ". The authors should clarify what this means for the reader as regards their definitions of S/LVOC, IVOC, and VOC.

Reviewer #2 (Remarks to the Author):

The paper describes the OA trend in China from 2013 to 2020. They used observational network data (2017-2020) and field campaign data (2013-2020) to suggest that OA decrease in China is probably due to the decrease of POA, to a lesser extent by SOA. They further use a chemical transport model and was able to reproduce this trend of OA. The work is interesting and has implications on future emission control as well as emission control strategies in developing countries. However, I find the paper lack of clarity on a number of things. I have several comments:

1. The paper does not provide a mechanistic explanation about why SOA does not decrease as much as POA (Figure 1c-f). This could be a major highlight on this paper. The authors mention the potential role of radiation and atmospheric oxidation but there is little clarity on how exactly this could play a role (Line 159-188). If their model gets this trend right (Figure 3), they should be able to provide some insights at least from the model simulations rather than mere speculation.

This also makes me wonder about Figure 4. If the trend during 2013-2020 is mainly driven by residential sources (as stated in the abstract) and is mainly due to the decrease in POA, why Figure 4 shows similar reduction in POA and SOA from 50% and 100% cut in residential. Am I missing anything here?

2. It is important to understand the OA trend in different seasons. The annual mean, as used in this study, can be heavily weighted towards winter when PM_{2.5} is higher, without much insights on other seasons (Figure S14). For Figure 1A-B, if annual mean has decreased a lot, it is expected that PM_{2.5} has decreased more in winter than in summer, if the decrease is driven by residential fuel burning (Figure S1

suggests that from emissions).

It would be great to show the seasons of each campaign in Figure 1c-f, as strong seasonality is expected for field campaign data. The authors could consider divide all those field campaigns into different seasons (Figure 1c-f), and examine the trends in each season, which would provide a much clearer message than altogether.

3. It should be noted that the main conclusion in this paper is actually drawn from all the field campaign data over the period of 2013-2020. The national network only covers the period of 2017-2020. Some careful analysis on the field campaign is indeed necessary.

4. Is the model validated by observations? The model does not reproduce the trend in YRD (Figure S4). Any reason for this?

5. Can the author explain how much difference the population weighting can make on OA concentrations? Do they consider the population weighting on the model trend too?

Response to reviews

We thank the reviewers for their constructive comments. In the following section, reviewer comments are in black. Point-by-point responses are in blue labeled with [R]. Line numbers in the responses correspond to those in the track-change version of the revised manuscript. Specific modifications to the manuscript are in red.

Reviewer 1

This study investigates the long-term trends of organic aerosol (OA) in China, motivated by the Chinese government's two-phase program to reduce OA between 2013-2020. The researchers combine modeling and observation methods effectively. Nation-wide observations are gathered from the North China Plain and province-level sites in the Yangzi River Delta. Aerosol mass spectrometers (AMS) and aerosol chemical speciation monitors are used for data collection. The authors employ the GEOS-Chem chemical transport model in high-resolution nested-grid mode. Various configurations, including simulations with 2014-2020 emissions and fixed 2013 meteorology, enable separation of OA precursor reduction effects from meteorological influences on OA concentrations. Notably, the study explores the implications of the fixed-emissions model runs using a random forest model, identifying meteorological variables most important to OA variability, an interesting approach.

Key findings include:

1. Declines in OA are mainly due to decreased Primary OA (POA).
2. Anthropogenic emissions drive OA trends, especially for POA.
3. Meteorology significantly affects Secondary OA (SOA), with $\sim\pm 5\%$ variations due to meteorological factors.
4. Meteorological features show negative correlations of both POA and SOA with precipitation, humidity, and non-stagnant days. Cloud-related factors impact SOA more than POA due to their influence on photochemistry.
5. Reductions in SO_x and NO_x in China offset each other, resulting in negligible net changes in annual mean POA and SOA.

To further reduce OA pollution, the authors recommend:

1. Prioritize replacing solid fuels with clean fuels or electricity for residential fuel burning.
2. Encourage products with less volatile components and focus on collecting and replacing high-emission products to reduce solvent-related SOA.
3. Continue the clean energy transition and invest in new technologies.

In sum, the authors have produced an excellent piece. This paper contains discussion of a novel data record, some elegant modeling techniques, and an investigation of the success of the last decade of Chinese air quality policy. It is a worthy achievement. The findings themselves are not a surprise: the finding that POA is easier to control than SOA is consistent with the US and European experiences; it is similarly unsurprising to learn that meteorological variability is important to SOA variability. The investigation of meteorological drivers is also as predicted: OA is anti-correlated with ventilation, precipitation, and relative humidity; SOA is anti-correlated with cloud fraction and optical thickness due to photochemical impacts. The surprise to me was that the SO_x and NO_x reductions offset one another, but the authors do not highlight these results. The work is valuable in that it is an examination of all of these findings in the Chinese context, but the findings themselves are as expected.

[R0] We sincerely appreciate the time and effort that Reviewer 1 dedicated to providing such insightful feedbacks on our manuscript. We provide point-by-point responses below and have revised the manuscript and supporting information (SI) accordingly. To quantify the offset effects of chemical drivers (in particular, SO₂ vs. NO_x), we performed further model analysis (Table S8: CTL_OTR_woSO₂ and CTL_OTR_woNO_x simulations). The results are presented in the updated text in Line 207-223 and 277-281 and Section S1 (Model scenarios) in SI with a new figure of Figure S15. We believe that the revised manuscript is significantly strengthened on the delivery and validation of findings, conclusions, and implications and provides a useful reference for developing countries.

Figure S15. The changes of SOA concentrations led by chemical drivers during the two action periods. The percent values in black represent the changes of population-weighted SOA concentration relative to 2013. The colored percent values in the lower right of each panel represent the contributions of emission controls on SO₂, NO_x, and other non-OA-precursor pollutants to the SOA changes. We summed up the changes of SOA tracers related to gas-phase oxidation processes to represent the impacts of oxidant levels and the changes of SOA tracers related to heterogeneous uptake processes to represent the impacts of ALWC as well as aerosol pH.

General issues: Model vs AMS definitions of POA and SOA

This paper primarily concerns a comparison between AMS observations and models of organic aerosol, to better understand the controls on the trends of these pollutants. These authors, like many before them, must confront an issue in making these comparisons: interpretations of AMS data do not make the same distinction between ‘primary’ and ‘secondary’ organic aerosols made by models. AMS observations are essentially apportioned between ‘oxidized’ and ‘not-oxidized’ OA, which is related to the POA/SOA distinction used by modelers but not identical to it.

The choices made by the authors in this comparison are logical: Hydrocarbon-like (HOA), cooking-related (COA), biomass-burning-related (BBOA) and coal-combustion-related OA (CCOA) is apportioned as POA, oxygenated OA factors – presumably some combination of Less-Oxidized Oxidized

Organic Aerosol (LO-OOA) and More-Oxidized Oxidized Organic Aerosol (MO-OOA) – are apportioned as SOA. However, they do not agree with several different recent studies of Chinese OA (Brewer et al., 2023; Wang et al., 2021). In both of those studies, the authors found that their observations were much more consistent with a treatment of POA which apportioned GEOS-Chem’s Emitted POA (EPOA) tracers to the HOA/COA/BBOA/CCOA and oxidized POA (OPOA) tracers to the OOA alongside SOA. This disagreement actually presents an opportunity for the authors here to explain how their model and their finding fit into the existing literature.

It is not clear to me how the authors differentiated between what GEOS-Chem Classic labels EPOA and OPOA, though an examination of the cited paper by Miao et al (2021) [citation 19] seems to suggest that some portion of what would have been considered POA in the classic GEOS-Chem ‘simple’ and ‘complex’ mechanisms might now be considered SOA by the authors’ model. This would be consistent with their finding that the S/LVOC and IVOC together contribute ~75% of the simulated SOA in China (Lines 123-124), as I believe older GEOS-Chem mechanisms (Hodzic et al., 2016; Pye et al., 2010) mostly consider S/LVOC and IVOC as POA-forming. Regardless, I think the authors ought to address this in the text or supplement.

[R1] We agree with the reviewer that AMS-derived POA and SOA are not identical to the modeled ones. In this study, we think the comparisons between the sum of HOA, CCOA, COA, and BBOA to the modeled POA and the sum of OOAs to the modeled SOA are meaningful for the following reasons.

- (1) The four primary PMF factors show mass spectra and mass ratios of organic matter to organic carbon (OM:OC) similar to freshly emitted OA from transportation, residential or industrial combustion, cooking, and biomass burning at ambient loading levels [Aiken et al., 2008; Zheng et al., 2020], which covers the main sources of primary emissions in China.

The model in this study considered full-volatility-range organic emissions from all these primary sources except cooking and applied volatility-based partitioning to determine the portion of organics (S/LVOC) that can remain in the particle phase after being emitted (so-called the modeled POA; labeled as the EPOA tracer). An OM:OC ratio of 1.4 is used for EPOA which is consistent with the ratios of PMF primary factors (1.2-1.6). In this regard, we think it is reasonable to use the comparison between the sum of PMF primary factors to the modeled POA as a validation of the model settings of the S/LVOC emissions and their volatility distributions as long as cooking is not a major contributor.

Cooking emissions are lack of good constraints on a national level. The PMF results indicate 8-33% of the OA mass as COA with insignificant yearly changes in cities [Zheng et al., 2023]. We expect <4% of the populated-weighted OA mass from cooking by assuming a COA mass fraction of 15% in model grids with population densities of >1000/km² and little interannual variations. Therefore, the overlooked COA should not affect much the modeled POA mass and trends herein. We kept COA in the observation dataset because the PMF analysis has been proven to robustly separate primary factors from oxidized “secondary” factors while individual primary factors may not be well resolved depending on instrument tuning and analysis skills. Nevertheless, for the model-observation comparison of POA in Fig. S4C, if COA is excluded from the observation data, the comparison gives a r of 0.85 and a NMB of 0.25, still supporting a good model representation of S/LVOC.

Other evidence to support the good model representation of S/LVOC include the consistent China’s emissions with other inventories (Fig. S3), the consistent estimation of US emissions with the findings of Pye et al. (<https://doi.org/10.23719/1527956>) (Fig. S4), and the roughly reproduced total mass

fraction of volatility bins below C^* of $1 \mu\text{g m}^{-3}$ (predominantly in the particle phase) although the fraction of individual bin has not been well captured (Fig. S21).

- (2) The PMF OOA factors are identified by mass spectra similar to SOA and much greater OM:OC ratios (1.8-2.2) than the ratios of primary factors (e.g., [Zheng *et al.*, 2020]). These OOA factors may represent SOA from various precursors with different photochemical age as well as aqueous SOA [Xu *et al.*, 2017; Zheng *et al.*, 2023]. In the model, SOA is the sum of biogenic SOA, aromatic SOA, IVOC SOA, S/LVOC SOA (labeled as the OPOA tracer), and key aqueous SOA, covering almost all important precursors as well as the oxidation and aqueous processes. An OM:OC ratio of 2.1 is applied to the modeled SOA, which is again consistent with the PMF findings. We think the comparisons between the sum of PMF OOAs to the modeled SOA along with the model-observation comparison of precursor concentrations validate general model skills on the SOA precursor emissions and chemical processes.

Indeed, we are aware of the confusing model treatments of primary emissions and the attribution of tracers when comparing to the AMS results among studies. To address that, we added a new table in SI (Table S9) to summarize the global burdens of OA reported in different GEOS-Chem studies.

Table S9. Global burdens of OA under different model schemes.

Source	Category	Tracer or Precursor	Global burden [Tg]				
			Simple scheme [Pai et al. , 2020]	Complex scheme [Pye et al. , 2010] ^a	Hodzic scheme ^b [Hodzic et al. , 2016]	Complex scheme [Pye et al. , 2010]	Revised complex scheme [This study]
			Non-volatile POA treatment			Semivolatile POA treatment	
			OM:OC=1.4 for EPOA and 2.1 for OPOA ^c			two-product surrogates	5-bin volatility distributions
			2013	2000	2005-2008 mean	2000	2013 ^d
	POA		0.84	0.92	0.94	0.03	0.32
Anthropogenic and biomass burning	SOA	EPOA	0.06	--	--	0.03	0.32
		OPOA	0.78	--	--	Not applicable	Not applicable
		S/LVOC	-- ^e			0.81 ^g	0.58 ^g
		IVOC	--	Not included	0.21	0.09	0.52
		Aromatics	--	--	0.08	Not specified	0.03
	OA_{anth/bb}		1.53	--	1.23	(~1.0) ^f	1.45
All except marine	OA		1.86	1.64	1.82	1.65	1.90
	Model results are evaluated by	Aircraft OA data			IMPROVE (US) and EMEP (Europe) OC data; AMS OOA data for rural/background; Aircraft OA data		NCP and YRD OC data in China; AMS PMF (POA and OOA) data; Precursor concentrations and volatility distributions

^a Pai *et al.* [2020] also reported 2013 burden for complex scheme. Their burden values are greater than those reported by Pye *et al.* [2010] when the primary OC emissions in the two studies are similar (31.2 vs 29.0 Tg C/yr). We list only the values from Pye *et al.* [2010] in the table.

^b The ND_DPH setting is considered as the Hodzic scheme used by Brew *et al.* [2023]. Brew *et al.* [2023] further improved aromatic SOA parameterization.

^c The OM:OC ratio is 2.1 for both EPOA and OPOA in [Pye *et al.*, 2010] while the OM:OC ratios are not reported in [Hodzic *et al.*, 2016].

^d The burden is calculated on the default configuration of GEOS-Chem (v13.3.1) with the revised complex scheme for OA.

^e Included in the calculation but the values are not specified in the literature.

^f The value is not specified in the literature. The aromatic SOA burden should be less than 0.10 Tg.

^g Labeled as the OPOA tracer.

As listed in Table S9, the modeled POA under non-volatile POA treatment are much greater than the modeled POA under semi-volatile treatment. This is because the quartz filter based POA emissions represent organic emissions over a range of volatilities. Some semi-volatile portion of this emission remain in the particle phase during the atmospheric dilution but some are gas precursors that may further produce SOA (S/LVOC-SOA). Advanced emission profile measurements suggest a scaling factor of 1.0

(1.4 for gasoline) to the filter-based POA emissions to represent the total (gas+particle) emissions of S/LVOC of anthropogenic sources [Lu *et al.*, 2020; Lu *et al.*, 2018]. Therefore, the non-volatile POA treatment overestimates the particulate organic emissions. This is confirmed by significant greater modeled POA than the AMS-derived POA in previous studies [Brewer *et al.*, 2023; Jo *et al.*, 2013].

The question is what fraction should be positioned as freshly emitted POA. Brew *et al.* [2023] found EPOA under non-volatile POA treatment reproduce better the AMS-derived POA, and the rest (the hydrophilic portion of non-volatile POA; also labeled as OPOA), if considered as additional S/LVOC-SOA, leads to a match between modeled SOA and AMS-derived OOA. This indicates that a majority of the filter-based POA emissions should be gaseous S/LVOC in the “KORUS-AQ” region [Nault *et al.*, 2018] and that the Hodzic representation on S/LVOC- and IVOC-SOA is insufficient.

In this study, we developed full-volatility-range organic emissions. The S/LVOC-SOA and IVOC-SOA are parameterized on the volatility distributions of their precursor emissions (8 bins total). With the revised model scheme, a high fraction of the filter-based POA emissions remain as EPOA in highly polluted anthropogenic source regions as shown in Figure S23, which explains why the model can reproduce high concentrations of AMS-derived POA in China. The EPOA/(EPOA+OPOA) fractions decrease to <0.4 in KORUS-AQ-related model grids. Note that EPOA/(EPOA+OPOA) needs to be more than 0.1 [Brew *et al.* 2023]. Table S9 suggest that S/LVOC-SOA+IVOC-SOA in the revised scheme is comparable to OPOA+SVOC-SOA+IVOC-SOA in the Hodzic scheme with non-volatile POA treatment, indicating a consistent SOA budget between our study and Brew *et al.* [2023].

Figure S23. The modeled EPOA / (EPOA + OPOA) at (A) surface and (B) at the altitude of 3 km in 2013. EPOA and OPOA are the tracers that represent the modeled POA and S/LVOC-SOA, respectively.

To sum up, previous studies might have misplaced some categories of OA and the standard complex scheme is outdated. On the total amount of OA, our study is consistent with previous model studies that were evaluated extensively by the observations [Hodzic *et al.*, 2016; Pai *et al.*, 2020]. We provide a bottom-up volatility-based simulation scheme to capture the underestimated anthropogenic contributions with careful evaluations for China. Our treatment of S/LVOC-SOA (although labeled as OPOA in this study and IVOC-SOA as the modeled SOA is constrained by the AMS-derived OOA and ambient flow-tube observations on SOA potential [Hu *et al.*, 2022]. Globally, well-constrained emission inventories over the entire volatility range along with improved representation of their SOA formation are needed to reproduce the spatial variabilities of OA.

We have added the above discussion and references in Section S3 in SI and revised the text in Line 149, 284-307, 417.

Specific Edits:

Line 105 – I was surprised by the ‘35% of NMVOC acting as an OA precursor’ value, and went and looked through the cited work by Li et al. (2014). I did not find the source of the claimed percentage. It is entirely possible that I have missed something, and welcome being corrected, but the authors should double check the citation.

[R2] Thanks for pointing out this confusion. Li et al. (2014) reported 14% of NMVOC emissions in MEIC as aromatics for the year of 2006, and Li et al. (2019) reported 33% as aromatics for 2017. The large increase is attributed to a rapid expansion of industrial and solvent use activities since 2000 (Li et al., 2019). We also use the MEIC inventory for NMVOC but with updates on the residential biofuel and coal use. The NMVOC emissions in this study are slightly different from those reported by Li et al. (2019) for the same years. To clarify, we have made several updates as follows: ① Line 124-127 “about 35% of non-methane VOC emissions (mostly aromatic compounds) may act as OA precursors (i.e., 9.0-10.3 Tg y^{-1})²². The high aromatic fraction in China is explained by the rapid increase of solvent use and industrial activities since 2000 (Li et al., 2019)”, ② Fig. 2’s figure caption “(A) Anthropogenic emissions of OA precursors from different source sectors. Among VOC, only aromatic compounds, glyoxal, methylglyoxal, isoprene, and terpenes are considered as OA precursors”, ③ Fig. S1’s figure caption “(B–D) The annual mean emission compositions of ~~total-NMVOC (include non-OA precursors)~~, IVOC, and S/LVOC grouped by species or C^* at 298 K in China. Aromatic compounds, GLY, MGLY, isoprene, and terpenes are typical OA precursors, which are about 34-37% of NMVOC and are outlined by light green wedges in panel B. In panel C, the light green wedges highlight the volatility bins that have the lowest vapor pressure and highest SOA formation potential. In panel D, the light green wedges highlight the volatility bins that are primarily distributed in the particle phase”, and ④ a few words in Section S1 (NMVOC emissions and VOC-type OA precursors) in SI.

Figure 3: I really like this figure. It is both information-dense and understandable. The presentation choices in panels A-D in particular are very well done. My only complaint is with Panel F. This panel contains several confusing references: the “Conc.”, “TROPO-P”, and “P1” labels should be clarified. Two of these features (“Conc.” and “TROPO-P”) do not appear to be discussed in the supplemental materials.

[R3] We have updated Figure 3E-F with a better description of the parameters.

Moreover, the figure caption for Fig. 3E-F is revised as follows: “CONC stands for the concentrations of POA and SOA used in RF which are the model outputs under the fixed-2013-emission scenario. PBLH, SWR, T(2m), U(10m), and V(10m) represent planetary boundary layer height, surface shortwave radiation, air temperature at 2 m, the east-west and north-south wind components at 10 m, respectively. The subscripts, 1 and tp, denote the first model layer and tropopause, respectively”. We have also revised Sect. S1 (Random forest model and SHAP analysis) and Fig. S13 in SI with the same set of labels.

Line 172: inclusion of “Besides” here is confusing and out of place.

[R4] This sentence is revised as follows: “~~Besides~~, Two other parameters, specific humidity (SH) and friction velocity (U^*), are more important for SOA than for POA”.

Line 194: “Except the uncertainty in precursor emissions, the representation of chemical processes for the SOA formation and aging is greatly simplified in chemical transport models.” This line is confusing to me. It seems to suggest that CTMs don’t simplify the uncertainty in precursor emissions, but this is not very true, especially considering the assertion in lines 123-124 that S/LVOC and IVOC contribute ~75% of SOA in China. Possibly one could argue that the representation of precursor emissions is only ‘very simplified’ rather than ‘greatly simplified’ but I’m not sure that’s a worthwhile line to draw. I suggest rephrasing this line to make the intended statement clearer.

[R5] We have rephrased this part as follows: “~~Except the uncertainty in precursor emissions, the~~ The representation of chemical processes for SOA formation and aging in chemical transport models can be an important source of model bias⁶. Our model simulations reasonably capture the regional variations of OA (Fig. S17 and Table S2), suggesting minor impacts of simplified model representation of multigenerational production and fragmentation pathways on SOA in highly polluted environments in China”.

Line 228-229: authors suggest that the large decrease in OA over China from 2013-2020 is “unprecedented” (a typo) and then in the very next sentence suggest a precedent: “[a] similar extent of decline ... occurred in the United States from 1990-2012”. I think this is the wrong word choice.

[R6] Agreed. We have revised the statement as follows: “Our study identifies a large and rapid decrease in OA from 2013 to 2020 in China from observations and model perspectives, indicating remarkable ~~unprecedented~~ achievements of the implementation of clean air actions (especially residential fuel replacement). A similar extent of decline (25%-50% of OC) occurred in the United States from 1990 to 2012”.

Line 242: “solvent use is the second large contributor” should read “solvent use is the second largest contributor”.

[R7] We have revised the text as suggested.

Line 246-247: “lowering OA mass ... is essentially a thrust on clean energy transition and technology innovations.” This phrasing doesn’t make sense – I don’t know what the authors mean by ‘a thrust’.

[R8] We have deleted this sentence and added three new policy implications (see [R9]).

Lines 249-260: This section is included in the ‘Policy Implications’ section of the paper but does not really consist of any policy implications. It instead lays out two (extremely correct) needs for future research, which I summarize as:

1. Characterization of OA species (or OA precursor emissions class or emissions sector) by health impacts is urgently needed to better tailor control strategies at the most damaging OA sources.
2. More research is necessary on potential future climate impacts on OA, and how that will change policy.

These points are excellent but not policy. I assume this inclusion was made because the paper contains no ‘Conclusions’ section, which is where they really do belong. The authors should re-organize this section to make clear exactly what the policy recommendations are and what the attendant research is that could make those recommendations even better. I suggest a numbered list of the three real policy implications in this section, followed by this additional discussion of necessary future directions.

[R9] Great point! We have re-organized the entire section and changed the section title as “**Discussion**” so that uncertainties, implications and future research needs all fit in. We followed the reviewer’s suggestion and made a number list of three policy implications and discussed two research directions as follows in Line 308-343: “Our findings have three important policy implications for OA pollution control: (1) The residential sector is still the biggest contributor of OA in China. As the residential fuel replacement slows down, the OA pollution likely becomes persistently high in China and thus challenges the consideration of tightening national air quality standards. Future policies on air-quality improvement should prioritize the continuation of residential clean-energy transition; (2) Abatement of VOC has been suggested as a near-future priority for ozone-pollution control in China 31. Solvent use is now the largest contributor of NMVOC and VOC-IVOC SOA precursors. Coordinated control on ozone and SOA should encourage measures on solvent-use emissions; (3) Our model simulations suggest the NO_x reduction increased SOA during 2013-2020 in China. In longer term, if the NO_x-suppressed regime is transitioned to a NO_x-limited regime, the NO_x reduction would decrease surface OH (lowering photochemical production) and inorganic nitrate concentrations (lowering ALWC and thus aqueous SOA), leading to co-benefits on SOA pollution. Notably, high emissions of OA precursors are expected in other emerging economies such as India and Southeast Asian countries (Fig. S18). High concentrations of OA in PM_{2.5} have already been a serious problem there. Similar considerations may be taken to develop effective pollution control measures in those countries. To gain maximum co-benefits and ensure the effectiveness of pollution mitigation policies, future research needs to focus on (1) characterization of OA species or OA precursor class or emission sectors by health impacts, which leads to better tailored control strategies at the most damaging OA sources; and (2) quantification of the potential climate impacts on OA under different future emission scenarios.”

Line 275: “24% of the emitted carbon present in low-volatility bins with saturation concentrations (C*) of $\leq 1 \text{ ug m}^{-3}$ ”. The authors should clarify what this means for the reader as regards their definitions of S/LVOC, IVOC, and VOC.

[R10] We meant to highlight the POA portion would be quite large under conditions of high OA concentrations in China when 24% of the S/LVOC emissions distribute in low-volatility bins. However, we noticed such information has been already delivered in Line 117 and therefore deleted this sentence from Methods.

References:

- Miao, R., Chen, Q., Shrivastava, M., Chen, Y., Zhang, L., Hu, J., et al. (2021). Process-based and observation-constrained SOA simulations in China: the role of semivolatile and intermediate-volatility organic compounds and OH levels. *Atmospheric Chemistry and Physics*, 21(21), 16183–16201. <https://doi.org/10.5194/acp-21-16183-2021>.
- Pye, H. O. T., Chan, A. W. H., Barkley, M. P., & Seinfeld, J. H. (2010). Global modeling of organic aerosol: the importance of reactive nitrogen (NO_x and NO₃). *Atmospheric Chemistry and Physics*, 10(22), 11261–11276. <https://doi.org/10.5194/acp-10-11261-2010>
- Wang, J., Ye, J., Zhang, Q., Zhao, J., Wu, Y., Li, J., et al. (2021). Aqueous production of secondary organic aerosol from fossil-fuel emissions in winter Beijing haze. *Proceedings of the National Academy of Sciences*, 118(8), e2022179118. <https://doi.org/10.1073/pnas.2022179118>.
- Li, M., Zhang, Q., Streets, D. G., He, K. B., Cheng, Y. F., Emmons, L. K., Huo, H., Kang, S. C., Lu, Z., Shao, M., Su, H., Yu, X., and Zhang, Y.: Mapping Asian anthropogenic emissions of non-methane volatile organic compounds to multiple chemical mechanisms, *Atmos. Chem. Phys.*, 14, 5617–5638, <https://doi.org/10.5194/acp-14-5617-2014>, 2014.
- Aiken, A. C., et al. (2008), O/C and OM/OC ratios of primary, secondary, and ambient organic aerosols with high-resolution time-of-flight aerosol mass spectrometry, *Environ. Sci. Technol.*, 42(12), 4478-4485, doi:10.1021/es703009q.
- Brewer, J. F., et al. (2023), A Scheme for Representing Aromatic Secondary Organic Aerosols in Chemical Transport Models: Application to Source Attribution of Organic Aerosols Over South Korea During the KORUS-AQ Campaign, *J. Geophys. Res.*, 128(8), doi:10.1029/2022jd037257.
- Hodzic, A., P. S. Kasibhatla, D. S. Jo, C. D. Cappa, J. L. Jimenez, S. Madronich, and R. J. Park (2016), Rethinking the global secondary organic aerosol (SOA) budget: stronger production, faster removal, shorter lifetime, *Atmos. Chem. Phys.*, 16(12), 7917-7941, doi:10.5194/acp-16-7917-2016.
- Hu, W., et al. (2022), Oxidation flow reactor results in a Chinese megacity emphasize the important contribution of S/IVOCs to ambient SOA formation, *Environ. Sci. Technol.*, 56(11), 6880-6893, doi:10.1021/acs.est.1c03155.
- Jo, D. S., R. J. Park, M. J. Kim, and D. V. Spracklen (2013), Effects of chemical aging on global secondary organic aerosol using the volatility basis set approach, *Atmos. Environ.*, 81, 230-244, doi:10.1016/j.atmosenv.2013.08.055.
- Lu, Q., B. N. Murphy, M. M. Qin, P. Adams, Y. L. Zhao, H. O. T. Pye, C. Efstathiou, C. Allen, and A. L. Robinson (2020), Simulation of organic aerosol formation during the CalNex study: updated mobile emissions and secondary organic aerosol parameterization for intermediate-volatility organic compounds, *Atmos. Chem. Phys.*, 20(7), 4313-4332, doi:10.5194/acp-20-4313-2020.
- Lu, Q., Y. Zhao, and A. L. Robinson (2018), Comprehensive organic emission profiles for gasoline, diesel, and gas-turbine engines including intermediate and semi-volatile organic compound emissions, *Atmos. Chem. Phys.*, 18(23), 17637-17654, doi:10.5194/acp-18-17637-2018.
- Nault, B. A., et al. (2018), Secondary organic aerosol production from local emissions dominates the organic aerosol budget over Seoul, South Korea, during KORUS-AQ, *Atmos. Chem. Phys.*, 18(24), 17769-17800, doi:10.5194/acp-18-17769-2018.
- Pai, S. J., et al. (2020), An evaluation of global organic aerosol schemes using airborne observations, *Atmos. Chem. Phys.*, 20(5), 2637-2665, doi:10.5194/acp-20-2637-2020.
- Xu, W., et al. (2017), Effects of Aqueous-Phase and Photochemical Processing on Secondary Organic Aerosol Formation and

Evolution in Beijing, China, *Environ. Sci. Technol.*, 51(2), 762-770, doi:10.1021/acs.est.6b04498.

Zheng, Y., et al. (2020), Characterization of anthropogenic organic aerosols by TOF-ACSM with the new capture vaporizer, *Atmos. Meas. Tech.*, 13(5), 2457-2472, doi:10.5194/amt-13-2457-2020.

Zheng, Y., et al. (2023), Secondary formation of submicron and supermicron organic and inorganic aerosols in a highly polluted urban area, *Journal of Geophysical Research: Atmospheres*, 128(4), e2022JD037865, doi:10.1029/2022JD037865.

The paper describes the OA trend in China from 2013 to 2020. They used observational network data (2017-2020) and field campaign data (2013-2020) to suggest that OA decrease in China is probably due to the decrease of POA, to a lesser extent by SOA. They further use a chemical transport model and was able to reproduce this trend of OA. The work is interesting and has implications on future emission control as well as emission control strategies in developing countries.

[R0] We are grateful for the constructive comments provided by Reviewer 2. We provide point-by-point responses below and have revised the manuscript and supporting information (SI) accordingly. In the revised manuscript, we present new observation data to support the OA trends from 2013 to 2020 and the validation of the model. We have conducted seasonal analysis on the observation data and justified the results by seasonality. We believe that the revised manuscript is significantly strengthened with additional data analysis, providing a thorough investigation on the long-term trends of OA and the causes in China.

However, I find the paper lack of clarity on a number of things. I have several comments:

1. The paper does not provide a mechanistic explanation about why SOA does not decrease as much as POA (Figure 1c-f). This could be a major highlight on this paper. The authors mention the potential role of radiation and atmospheric oxidation but there is little clarity on how exactly this could play a role (Line 159-188). If their model gets this trend right (Figure 3), they should be able to provide some insights at least from the model simulations rather than mere speculation.

This also makes me wonder about Figure 4. If the trend during 2013-2020 is mainly driven by residential sources (as stated in the abstract) and is mainly due to the decrease in POA, why Figure 4 shows similar reduction in POA and SOA from 50% and 100% cut in residential. Am I missing anything here?

[R1] We thank the reviewer for the good suggestion. The lesser reduction in SOA than in POA is the result of the lack of control on solvent-use emissions of IVOC and VOC and the chemical and meteorological impacts. Residential fuel burning is the predominant source sector of POA. For SOA, residential and solvent use are the major contributors of the SOA precursors. During 2013-2020, the emission reduction is mainly from residential sector and solvent use emissions increased by 8%. Except difference in precursor emission reduction, the model results indicate a few percent of net increase in SOA led by chemical drivers and meteorology-driven changes of annual-mean SOA from -5% to 6%.

In Figure 4, 38% of OA is POA in the base case. Because the majority of S/LVOC is contributed by residential sources, a full elimination of residential organic emissions may greatly reduce POA by 63% and SOA by 38%, leading to a total decline of 41% of OA. The less reduction in SOA is expected because solvent use and industry contribute substantially to SOA except the contribution of residential sources. We have revised the text in Line 171, 207-223, 248-255, 270-283 to highlight the mechanistic explanation.

2. It is important to understand the OA trend in different seasons. The annual mean, as used in this study, can be heavily weighted towards winter when $PM_{2.5}$ is higher, without much insights on other seasons (Figure S14). For Figure 1A-B, if annual mean has decreased a lot, it is expected that $PM_{2.5}$ has decreased more in winter than in summer, if the decrease is driven by residential fuel burning (Figure S1 suggests that from emissions). It would be great to show the seasons of each campaign in Figure 1c-f, as strong seasonality is expected for field campaign data. The authors could consider divide all those field campaigns into different seasons (Figure 1c-f), and examine the trends in each season, which would provide a much clearer message than altogether.

[R2] We agree with the reviewer that seasonality needs to be considered more carefully in the analysis.

In the revised manuscript, we have examined the seasonal changes for both of the long-term OC and field-campaign OA data. As shown in the new figure below, seasonal mean OC concentrations in NCP are high in winter and low in summer as is consistent with the greater residential emissions of S/LVOC in winter. Indeed, the observed annual OC concentrations show a larger decrease by about 40% in winter than in other seasons (33%, 30%, and 13% for autumn, spring, and summer, respectively) from 2017 to 2021, explained by the greater wintertime residential emission reduction. Similar seasonal trends have been also observed at the research site in Beijing during 2013-2020.

Figure S1. The annual-mean OC concentrations in different seasons (DJF: winter; MAM: spring; JJA: summer; SON: fall) (A) in NCP obtained from the long-term continuous offline measurements at 34 sites of the national network and (B) in Beijing obtained from long-term online measurements at a research roof site in the campus of Peking University.

For field-campaign data, we replotted Figure 1. The revised Fig. 1C-D now shows the results in different cities in different seasons. The data in five cities from different regions in China indicate clear OA declines in winter, accompanied with increasing SOA mass fractions in three of the cities. In Beijing where the campaign data covers over 50% of the period of 2013-2020, the declines of OA and the increases of SOA fractions present in all seasons with more significant changes in winter.

Figure 1. Long-term observations of OA in China and their comparisons to model simulations. (A) Annual mean OC concentrations from continuous measurements at 34 sites in a recently-developed national monitoring network for PM_{2.5} chemical composition in North China Plain (NCP), 6 provincial

monitoring sites in Yangzi River Delta (YRD), and 3 research sites in the two regions. Error bars represent the standard deviations (NCP) and the ranges (YRD) of the annual concentrations across sites. (B) Annual mean OC concentrations at each site compared to the modeled OC. (C-D) Campaign-mean OA concentrations and SOA mass fractions in OA obtained from individual campaigns in different seasons (DJF: winter; MAM: spring; JJA: summer; SON: fall) from 2013 to 2020 in five cities in China (BJ: Beijing in NCP; SH: Shanghai in YRD; SZ: Shenzhen in PRD; LZ: Lanzhou in NW; XM: Xiamen). (E-F) Campaign-mean POA and SOA concentrations compared to the modeled concentrations. Sites are marked in the map.

To clarify the seasonal effects, we have revised the results and discussion in Line 85-97, 120, 154 and added the discussion on the new Figure S1 in Section S3 (Comparisons of OC).

3. It should be noted that the main conclusion in this paper is actually drawn from all the field campaign data over the period of 2013-2020. The national network only covers the period of 2017-2020. Some careful analysis on the field campaign is indeed necessary.

[R3] As described in the revised Section S2 (Long-term organic carbon measurements), we have added extra long-term OC measurement data from 4 more provincial monitoring sites and 9 research sites in the revised manuscript. The expanded datasets provide a better coverage for 2013-2017 as shown in the revised Fig. 1A, supporting the decline trend of OC over the whole action-plan period. For the field campaign data, we followed the reviewer's suggestion in comment #2 to examine the trends of OA, POA, and SOA in each season quantitatively in five cities from different regions in China. As shown in the revised Fig. 1C-D, the analysis is updated with the consideration of the seasonal variations. The text related to the observations (Line 75-107) has been revised accordingly.

4. Is the model validated by observations? The model does not reproduce the trend in YRD (Figure S4). Any reason for this?

[R4] Thanks for bringing up the important question. Yes, the model is validated extensively in this study as described in the revised Section S3. We first compared the modeled OC to the long-term measurements in NCP and YRD (Fig. S5C-D). The comparison to NCP network indicates a good model performance. For YRD, the original comparison only included two sites. Both sites are located in Zhejiang province. The model underestimated the OC concentrations. Two sites are too limited to evaluate the model performance. In the revised version, we expanded the long-term datasets to 6 sites (3 from Zhejiang province, 2 from Jiangsu province, and 1 from Shanghai). As shown in the new Fig. S5B, the model is able to reproduce the regional means of OC concentrations when more sites are included in YRD. We also added the scatter plots for model-site comparisons with more data from 9 research sites from the literature. The ones cover multiple years are shown in the revised Fig. 1A and the rest are single-year mean concentrations that we added to the comparisons in Fig. S5C-D. The model-observation comparisons show small NMB values for both NCP and YRD, indicating no significant bias of the model in a regional scale. The model has some overestimations in Jiangsu province and underestimations in Zhejiang province, indicating possible local emission biases. The comparisons to POA and SOA however do not show significant regional (Fig. 1E-F) and seasonal bias (Fig. S5E-F). As we discussed in Line 221-223, the model skills in describing finer temporal and spatial variations of OA are limited because of large uncertainties in emissions and the lack of sufficient observational constraints.

Overall, we think this study provides the best model simulations for OA in China with most comprehensive model validations on OC, POA, SOA, OA precursors and their volatility distributions as

well as OH levels. We have revised Figure 1, Figure S5, main text related to model validation and uncertainty (Line 50-53, 134-149, 284-307, 417-419), and Sections S2-S3 for clarification.

Figure S5 Observations compared to model simulations in China. (A-B) Annual-mean OC concentrations derived from continuous measurements at 34 sites in a recently-developed national monitoring network for PM_{2.5} chemical composition in NCP and at 6 provincial monitoring sites other sites in YRD. Only the sites with annual data coverage of >75% were used to calculate the annual-mean concentrations. In YRD, 1 or 2 sites did not meet the criteria for some years. The actual sites in each year are listed in gray in panel B. Error bars in panels A and B represent the standard deviations of the mean concentrations of sites in NCP and the ranges of the mean concentrations of sites in YRD, respectively. (C-D) Yearly averages of OC concentrations obtained from the NCP (n=34) and YRD (n=6) network sites (including low data-coverage ones as the model simulations matched with the measurement periods) as well as the literature data from different research sites in the two regions. (E-F) Mean concentrations of POA and SOA for from individual campaigns from 2013 to 2020 in China colored by different seasons (DJF: winter; MAM: spring; JJA: summer; SON: fall).

5. Can the author explain how much difference the population weighting can make on OA concentrations? Do they consider the population weighting on the model trend too?

[R5] As shown in the figure below, the trends of population-weighted and arithmetic-mean concentrations of POA and SOA in China are similar. The population-weighted POA and SOA annual mean

concentrations decrease by 54% and 25% from 2013 to 2020, while the arithmetic means decrease by 47% and 24%, respectively. The population-weighted concentrations are higher because high anthropogenic emissions usually occur in areas with high population density. In Fig. S5, we present the arithmetic annual-mean concentrations for OC and campaign-mean concentrations for OA for the purpose of comparisons. In Figs. 2-4, we present the population-weighted annual-mean concentrations to link the pollution levels to WHO PM_{2.5} guidelines directly. Using either of the results to present won't affect the conclusions and the discussion on the decline trends. We have added this figure as **Figure S22** and some text in Methods (**Line 393-396, 415-416**) and **Section S3 (Population weighting)** in SI for clarification.

Figure S22. The population-weighted and arithmetic-mean annual concentrations of POA and SOA in China.

REVIEWER COMMENTS

Reviewer #1 (Remarks to the Author):

The changes made in response to my comments were sufficient and the rebuttals convincing. However, the data is not really available to review at the provided Zenodo link - it contains neither the relevant GEOS-Chem code nor the actual dataset and code used to generate the analysis or figures in the paper.

Reviewer #1 (Remarks on code availability):

This is not really what I think of when I see someone discussing Code Availability. The authors have included some of the code of GEOS-Chem 13.3.1, something which is publicly available already. In point of fact, the authors do not actually include the source code for GEOS-Chem in their link - src/GEOS-Chem in the provided zip file is entirely empty. The portion of the code that would be most relevant to review - the author's OA scheme, which they discuss in the supplement and in their response to me - is not actually included in the file, and is not part of the standard GC 13.3.1 download available from the GEOS-Chem website.

The data I would most expect to be available here would be the model output datasets that the authors used to make their figures and perform their analysis, as well as the code used in that analysis and data presentation. However, it is not here and as such I cannot do any sort of analysis or replication on its contents. If code availability is a requirement for this submission, I do not believe it is in any way met by the zenodo link above (record/5703364). I am unable to use the provided code to make any meaningful statement on the quality or believability of the work.

Reviewer #2 (Remarks to the Author):

The authors have adequately addressed all my comments. I think the paper is ready to be published.

Response to reviews

We thank the reviewers for their comments. In the following section, reviewer comments are in black. Point-by-point responses are in blue labeled with [R]. Specific modifications to the manuscript are in red.

Reviewer #1 (Remarks to the Author): The changes made in response to my comments were sufficient and the rebuttals convincing. However, the data is not really available to review at the provided Zenodo link - it contains neither the relevant GEOS-Chem code nor the actual dataset and code used to generate the analysis or figures in the paper.

[R0]. We thank Reviewer #1 for the comments. We have updated the Data availability section as follows: “**The monitoring network observation data are available from the corresponding author upon request. The model outputs used in the analysis are available at <https://disk.pku.edu.cn/link/AA03DF14A7E715450BB628921DD9100516>**”. Note that all the campaign means of OA used in the analysis are listed in Table S1. We share the 2013-2020 monthly mean model outputs now in the public accessible folder. Those are the main model results that we used to produce the OA trends. Comparisons to site observations were calculated on hourly model outputs and the model results have been listed in Table S1.

Reviewer #1 (Remarks on code availability): This is not really what I think of when I see someone discussing Code Availability. The authors have included some of the code of GEOS-Chem 13.3.1, something which is publicly available already. In point of fact, the authors do not actually include the source code for GEOS-Chem in their link - src/GEOS-Chem in the provided zip file is entirely empty. The portion of the code that would be most relevant to review - the author's OA scheme, which they discuss in the supplement and in their response to me - is not actually included in the file, and is not part of the standard GC 13.3.1 download available from the GEOS-Chem website.

The data I would most expect to be available here would be the model output datasets that the authors used to make their figures and perform their analysis, as well as the code used in that analysis and data presentation. However, it is not here and as such I cannot do any sort of analysis or replication on its contents. If code availability is a requirement for this submission, I do not believe it is in any way met by the zenodo link above (record/5703364). I am unable to use the provided code to make any meaningful statement on the quality or believability of the work.

[R1] We have updated the Code availability section as follows: “**The revised GEOS-Chem code and the run directory are available at https://github.com/rqmiao/OA_simulation**”. Model installation and operation are well documented on the GEOS-Chem website. With the revised GEOS-Chem code, users are able to check our model updates which have been described in Sect. S1 and Tables S4-S8. They can also generate the executable file (“gclassic”) that we have already put in the uploaded run directory. With the run directory, users are able to reproduce our model outputs with all necessary input files. GC users often use GCPy: GEOS-Chem Python toolkit to analyze the model outputs (<https://gcpy.readthedocs.io/en/stable/index.html>). We have also provided an example of reading the model outputs in the run directory under the subfolder “example user code for analysis”.

Reviewer #2 (Remarks to the Author): The authors have adequately addressed all my comments. I think the paper is ready to be published.

[R0] We are grateful for the second-round review provided by Reviewer 2.

REVIEWERS' COMMENTS:

Reviewer #1 (Remarks to the Author):

The authors now provide the code required to analyze their GEOS-Chem setup, which is an improvement from the previous state. They have also provided the full GEOS-Chem output from their modeling runs. However, they have not included any of the code required to reproduce their figures. Overall, this represents an improvement from an 'open data' perspective, but still falls short of the CodeOcean setups that I have seen in other Nature Communications pieces. I have looked into some of their model output, but without the code used to generate the figures in question, it would take an enormous amount of time for me to regenerate their figures from scratch.

Reviewer #1 (Remarks on code availability):

See my response to the above. The authors have now provided a useful set of GEOS-Chem code, with which I can analyze the changes they have made to the GEOS-Chem SOA scheme. However, they have not included the code used to generate their figures and as a result I cannot opine on the decisions made in generating them without rebuilding the code myself entirely from scratch.

Response to reviews

Reviewer comments are in black. Point-by-point responses are in blue labeled with [R]. Specific modifications to the manuscript are in red.

Reviewer #1 (Remarks to the Author): The authors now provide the code required to analyze their GEOS-Chem setup, which is an improvement from the previous state. They have also provided the full GEOS-Chem output from their modeling runs. However, they have not included any of the code required to reproduce their figures. Overall, this represents an improvement from an 'open data' perspective, but still falls short of the CodeOcean setups that I have seen in other Nature Communications pieces. I have looked into some of their model output, but without the code used to generate the figures in question, it would take an enormous amount of time for me to regenerate their figures from scratch.

Reviewer #1 (Remarks on code availability): See my response to the above. The authors have now provided a useful set of GEOS-Chem code, with which I can analyze the changes they have made to the GEOS-Chem SOA scheme. However, they have not included the code used to generate their figures and as a result I cannot opine on the decisions made in generating them without rebuilding the code myself entirely from scratch.

[R0]. We thank Reviewer #1 for further suggestions. We are happy to provide the code to read the GEOS-Chem model output files, to produce statistical model results, and to generate the figures. The code are now added to the Github repository (<http://doi.org/10.5281/zenodo.11114855>) or (https://github.com/rqmiao/OA_simulation/tree/main/codeforfigures). Please note that some of the figure panels are plotted in spreadsheet-type software for which there are no code available (e.g., Fig. 1c-h).